# Micro- and Nano-Roughness Separation Based on Fractal Analysis

**DOI:** 10.3390/ma17020292

**Published:** 2024-01-06

**Authors:** Árpád Czifra, Erzsébet Ancza

**Affiliations:** 1Institute of Mechanical Engineering and Technology, Óbuda University, 1034 Budapest, Hungary; 2Institute for Natural Sciences and Basic Subjects, Óbuda University, 1034 Budapest, Hungary; ancza.erzsebet@bgk.uni-obuda.hu

**Keywords:** roughness, dominant wavelength, fractal, full-length scale, microtopography, nano-roughness, power spectral density

## Abstract

When describing the tribological behaviour of technical surfaces, the need for full-length scale microtopographic characterization often arises. The self-affine of surfaces and the characterisation of self-affine using a fractal dimension and its implantation into tribological models are commonly used. The goal of our present work was to determine the frequency range of fractal behaviour of surfaces by analysing the microtopographic measurements of an anodised aluminium brake plunger. We also wanted to know if bifractal and multifractal behaviour can be detected in real machine parts. As a result, we developed a new methodology for determining the fractal range boundaries to separate the nano- and micro-roughness. To reach our goals, we used an atomic force microscope (AFM) and a stylus instrument to obtain measurements in a wide frequency range (19 nm–3 mm). Power spectral density (PSD)-based fractal evaluation found that the examined surface could not be characterised by a single fractal dimension. A new method capable of separating nano- and micro-roughness has been developed for investigating multifractal behaviour. The presented procedure separates nano- and micro-roughness based on the geometric characteristics of surfaces. In this way, it becomes possible to specifically examine the relationship between the micro-geometry that can be measured in each wavelength range and the effects of cutting technology and the material structure that creates them.

## 1. Introduction

The separation of roughness and waviness has been included for decades to characterise surface microgeometry. The separation of unequal components of different wavelengths from different sources carries important information on the formation of each microgeometric element. They also play a different role in tribological processes, so the identification of each wavelength component is an important element of understanding friction, wear, and lubrication phenomena.

Different components of surface unevenness have different sources. The waviness of the surface bears a relation to the operating parameters of the production process, such as the vibration of the cutting machine. The micro-roughness is influenced by tools and production parameters of the surface finishing process, such as cutting tool geometry or cutting parameters. Submicro/nano roughness is associated with the material composition and microstructure of materials, for instance in the case of coatings.

Surface unevenness is one of the parameters that influence the behaviour of tribological processes. Different orders of surface roughness—depending on the hardness of the contacting materials—have different effects on real contact area [1] and wear resistance [2]. Many tribological phenomena are overviewed in [3] where surface nano- or micro-roughness has a significant effect, and authors introduce measuring techniques and their limitations in the frequency range. However, the usage of different measuring techniques is not enough for the real separation of micro- and nano-roughness based on physical grounds.

In roughness measurements, RC and Gaussian filters have traditionally been used to separate roughness and waviness. To meet growing demands, the enhanced dual-Gaussian filter and the robust or spline filters are available for industry and researchers [4,5,6]. The evaluation technique based on motif combinations, where there are also ways of separating roughness and waviness, has had long traditions [7]. In these technologies, the separation of roughness and waviness is not tied to physical characteristics but is based on a predetermined wavelength cutoff.

There are real needs behind which many methods have been developed to identify typical wavelengths. The German automotive standard introduced the so-called dominant wavelength concept [8] in the early 2000s, but morphological screening methods have also been revived [9,10]. Additionally, full-frequency spectrum experiments have had a greater role [8,10,11], which no longer involve identifying the frequencies but characterising the surfaces in the frequency space. These include wavelet-based characteristics, correlation functions, and Power Spectral Density (PSD) analysis based on the Fourier transform [12,13,14,15]. The most important advantage of these methods is that they provide true frequency separation based on the real geometric properties of surfaces.

Fractal analysis is a special area of study covering the entire wavelength/frequency spectrum. The PSD analysis is suitable for fractal characterization of surfaces [9]. The fractal dimension is a self-affine feature [16]. In the case of technical surfaces, self-affine has been proven [14,15]; however, the determination of the fractal dimension of surfaces, and especially the use of fractal characterization, has many contradictions [17,18]. At present, there is no unified standpoint and methodology that would accept the use of fractal technology to characterise technical surfaces, although many tribological methods—friction and hysteresis loss calculation theory [9,11,19,20,21]—build on the fractal nature of surfaces and require a full spectrum analysis, which the classical roughness parameters are unable to provide due to the relatively narrow frequency range of measurements.

The use of PSD analysis and fractal evaluation often leads to controversial results, thus its use in characterising technical surfaces requires caution. Our earlier studies [22] revealed the limitations of PSD-based evaluation techniques where the sampling step of the roughness measurement, the frequency step of the PSD curve, and the method of determining the Hurst exponent were included as critical points. In our subsequent investigations [23], we successfully applied the methodology to characterise the surfaces of different tribological behaviours.

The most important question about the self-similarity of surfaces is whether this self-similarity (self-affinity) can be extended to the entire wavelength spectrum. Our measurements cover only a narrow wavelength range (the lower limit is the sampling distance and the upper limit is the measurement range). The question is, can the self-similarity determined from these limited measurements be extended to the full wavelength spectrum? Bhushan’s examinations treated surfaces as bifractal as early as 1992 [24], but only the frequency ranges below and above the dominant wavelengths were separated. In recent years, however, several authors—such as Wu [18], Le Gal [11], and Ţălua [25]—have applied the terms bifractal and multifractal, but in each case, they interpreted concepts in a wavelength range smaller than the dominant wavelength. However, methodologically, it is not clear how individual fractal ranges can be separated.

The results of current research ([26,27,28]) show that the wavelength parameters of surfaces influence tribological processes.

Research [29,30,31] extends to the correlation between bifractal behaviour and tribological (contact or wear) properties. Scientists use the term bifractal to define two fractal dimensions, although the separation of the regimes is not exact.

In the course of this research, microgeometrical and microtopographic examinations of a piston of a brake master cylinder were performed using a stylus instrument and an atomic force microscope. The experiments were conducted at different sampling distances and different measuring lengths such that the measurements covered five orders of magnitude (19.6 nm–3 mm). 

The objectives of the tests were as follows: to determine the largest dominant wavelength of the surface using three independent methods. The techniques used were: standard 2D roughness parameters (1), 2D PSD analysis performed on roughness profile (2), and slicing technique performed on 3D topography to identify roughness peak (3).to examine whether the fractal dimension values characterising the surfaces are the same in the case of 3D topographical measurements in different wavelength ranges, i.e., whether the bifractal and multifractal behaviours mentioned in the literature can be detected. There is no clear position in the literature in this regard. To determine the fractal dimension, a PSD analysis was performed on 3D topographic measurements.to determine the limit wavelength of each fractal domain in the case of different fractal dimension values (bifractal or multifractal behaviour). The literature currently does not provide a solution to this problem.

## 2. Materials and Methods

### 2.1. Measurement of Brake Plunger

The brake system is one of the most essential structures of automobiles. Our previous investigations [23] proved that the surface roughness of the plunger has a considerable effect on the friction force. It has also been confirmed that different wavelength components of the surface result in different tribological behaviours [32].

Two types of measuring systems were used to examine the surface roughness of a brake plunger. The goal was to provide wide-frequency scale investigations. Different measuring equipment, including a Mahr Perthometer Concept stylus (Mahr GmbH, Göttingen, Germany) instrument and a D3100 atomic force microscope (Bruker Corporation, Billerica, MA, USA), were used. The scheme of the profile and topographic measurement positions and a photo of the measured plunger can be seen in Figure 1.

The stylus instrument at the Bánki Donát Faculty of Mechanical and Safety Engineering, Óbuda University, had an FRW750 diamond cone stylus (Mahr GmbH, Göttingen, Germany) with a 5 μm peak radius and 90° peak angle, which provided higher wavelength scale measurements. The atomic force microscope (AFM) at the Chemical Research Centre of the Hungarian Academy of Sciences with IIIA controller and Dimension 1951e type scanner (Bruker Corporation, Billerica, MA, USA) was used to carry out the low wavelength scale measurements.

Measurements were taken using two primary plungers manufactured using the same technology (indicated with A and B). First, standard roughness measurements were taken on a Mahr Perthometer Concept stylus roughness measuring instrument on a measuring length of 5.6 mm, with 10 roughness profiles, and parallel to the longitudinal axis of the cylinder, on both specimens. Measurements were made in accordance with ISO 4287 standards [33]. This was followed by topographic measurements, for which the measurement settings are summarized in Table 1. Sampling distances were the same in both the x and y directions.

Based on the preliminary 2D tests, the planned 3D measurements of the A plunger were supplemented with 1-1 measurement: (3000 × 3000 µm^2^) tactile measurement and (90 × 90 µm^2^) AFM measurement. The aim of the former was to measure a surface area significantly larger than the assumed dominant wavelength, while in the latter case, we wanted to achieve a larger number of measurements in the range that is the immediate vicinity of the expected dominant wavelength.

### 2.2. Determination of the Dominant Wavelength

Three different methods were used to evaluate the measurements in order to obtain reliable dominant wavelength results.

To determine the dominant wavelength of the surface, the parameter *RSm* (mean width of profile elements), which is defined in ISO 4288 [34], was used first.

Equation (1) in which the interpretation of *Xsi* is shown in Figure 2 is used to determine the parameter.
(1)$$RSm=\frac{1}{m}\sum_{i=1}^{m}Xsi$$

Secondly, power spectral density analysis was performed to identify the dominant wavelength on the roughness profiles. The methodology used was described in our earlier work [23]. The following example of a turned profile (see Figure 3a) produced with a feed rate of 50 µm/revolution illustrates the application of the method. The tool marks have periodical character, but they differ from the common turning grooves. There are two possible ways of showing the results. One is to represent the amplitude of PSD in the wavelength function (see Figure 3b). The practical gain of the first representation is that dominant wavelength components appear as peak points on the PSD curve. Despite the unusual form of tool marks, the highest point on the PSD curve can be found at 50 µm, indicating that this is the dominant wavelength of the profile. The other prevalent method is the logarithmic scale frequency–PSD amplitude visualization (see Figure 3c). As can be seen, the high-frequency range of the curve can be approximated using a straight line. Below the q* frequency value, the curve can be approximated using a horizontal line (see [9,17,18,24]). The q* frequency corresponding to the breaking point gives the dominant wavelength of the surface. 

The third method for determining the dominant wavelength was based on 3D surface roughness measurements. The roughness peaks were identified using the slicing method on the roughness topographies (see methodology in Figure 4 and details in [35]).

In terms of our algorithm, an asperity can be defined as a geometric conformation displayed by a connected set of measurement points located over the mean plane, at the intersection of another parallel slicing plane. The height of the slicing plane was not fixed, but “n” number of slicing steps were performed over the mean plane, for which asperities were determined again and again. The slicing technique was applied to the largest measured topography (3 mm × 3 mm). We searched for the slicing height at which the highest number of roughness peaks could be identified during slicing. The slicing height showing the highest number of roughness peaks was 0.803 μm, where 135 asperities were identified. Next, we examined the roughness peaks identified this way. After asperity identification, the major axis was evaluated asperity by asperity as the length of the segment connecting the two points of the asperity contour located the farthest from each other in the x–y (slicing) plane. The minor axis was perpendicular to the major axis in the middle point thereof (see Figure 4).

Having completed the identification of the major and minor axis for the 135 roughness peaks, we determined the ratio of the major and minor axis in each case and then prepared the distribution curve of the axis ratios.

As a last step, the maximum of the distribution curve (RDC, see Figure 4) was determined as the axis ratio value of the topography. After that, the entire surface was covered with 135 RDC rectangles. The length of the longer side of the rectangles defined in this way was considered the dominant wavelength.

### 2.3. Fractal Analysis

3D topographic measurements were used to determine the fractal dimension. 3D PSD analysis was performed for the evaluation. Figure 5 represents the basic steps of the applied method and mathematical details can be found in [23]. 

In 3D PSD analysis, the frequency range depends on the measured area and sampling distance. The lowest frequency was the reciprocal of the measuring length, while the upper limit was based on the Nyquist–Shannon sampling theorem. This frequency range was divided into 125 parts in both coordinate directions and then, using a consolidation proposed by Persson [9], a 2D logarithmic scale PSD curve was generated. The basis of Persson’s reduction is that PSD topography seems like a truncated cone (see the upper right part of Figure 5) that needs only two coordinates (namely the height and the radius) to be described. The resulting 15,625-point PSD curve (see the lower right part of Figure 5) was used to determine the slope of the curve from which the fractal dimension can be determined.

The Hurst exponent (*H*) can be determined from the slope (*S*) of the straight line fitted to the PSD curve based on Equation (2):(2)$$H=\frac{-S}{2}-1$$

From this, the fractal dimension of the surface (*D_f_*) can be determined as:(3)$$D_{f}=3-H$$

The final step of the study was the interpretation of fractal dimension values in a frequency spectrum (see the lower left part of Figure 5). Since the measurements covered different frequency spectra, it was possible to assign the fractal dimensions defined in the different frequency ranges to the frequency band. In practice, this meant that the value of the fractal dimension of the given measurement was assigned to the mean value of the frequency range of each measurement, and the resulting points were plotted on a logarithmic frequency scale. No similar methodology was found in the literature. 

## 3. Results

### 3.1. Dominant Wavelength

The wavelength of measurements on roughness profiles is indicated by the parameter of the average distance of the profile elements (*RSm*). The average *RSm* parameter on the 10 measured profiles was 163.1 μm, with a deviation of 24.6 μm. A typical roughness profile of the surface is shown in Figure 6. The PSD function of the profiles is much more informative. Figure 7 shows the PSD amplitude dependent on the wavelength, of which the 178 μm value stands out, indicating the dominant wavelength of the surface.

The stylus microtopography of the surface can be seen in Figure 8 and Figure 9. Figure 8 shows a 1 mm × 1 mm surface area, while Figure 9 shows a 200 μm × 200 μm portion of this. Determining the dominant wavelength is not easy either here; as can be seen in the profile of Figure 6, the dominant elements on the surface have 2–4 local maxima. Note that the value of the *RS* parameter—an average distance of local maxima—was 32.9 μm on average measured profiles, i.e., significantly different from *RSm*. This effect can be detected in the PSD curve and the topographic image.

When applying the slicing technique, the interrelated set of points above the slicing planes are interpreted as roughness asperities, regardless of whether they have one or more local maxima. Thus, it is possible that this technique identified 135 asperities at the 3 mm × 3 mm topography at a height of 0.803 μm (the number of asperities identified when slicing below this level decreased as more and more surface areas were merged, while the number of asperities identified when slicing above this level also decreased due to cutting less material; that is, we only identified the peaks of asperities). Figure 10 shows the distribution function of the ratio of the major and minor axis of the identified asperities. The maximum of the curve is 1.7, which means slightly extended asperities. If the measurement area is covered with 135 identical rectangles with an aspect ratio of 1.7, the lengths of the rectangles would be 95 and 161 μm. 

The dominant wavelengths that can be determined using the three techniques are summarized in Table 2. Due to the fragmentation of the surface, smaller (35–40 μm) and larger (220 μm) dominant elements also occur.

It is important to note that the applied sampling step remains significantly below the dominant wavelength of the surface even in the case of the rarest scan (3 mm × 3 mm topography). This is a necessary condition for reliable fractal evaluation. It is also important to point out that the measurement range of AFM measurements (from 5 μm to 90 μm) is below the dominant wavelength, i.e., focuses only on some details of dominant microtopographic elements. In the case of the 90 × 90 μm^2^ AFM measurement, the dominant elements or their individual details can be easily identified, as is also shown in Figure 11.

### 3.2. Fractal Analysis

During fractal analysis of the surfaces, the breakpoint typical of the dominant wavelength, already mentioned by Bhushan, was still identifiable on the larger frequency ranges (3 × 3 and 1 × 1 mm^2^ surfaces). Figure 12a shows the 3 × 3 mm^2^ topography of the PSD topography, while Figure 12b shows the derived 2D PSD curve, which consists of 15,625 points. The black curve represents the smoothing of the PSD curve with a 200-point simple moving average. The end of the initial horizontal section is unclear, but the peak at *λ_0_* gives the dominant wavelength as 155 μm; this correlates well with previous results of profile analysis (163, 178, and 161 μm). The linear regression line can be seen in the highlighted blue part of Figure 12b. The fractal dimension in the given frequency range was determined from the slope of this straight line.

In the case of AFM measurements, where the lowest frequencies obtained are below the dominant frequency of the surface, the horizontal initial phase of the PSD curves is missing. This can be seen in Figure 13, which shows the analysis of the 25 μm × 25 μm surface area.

The results of the fractal dimension are summarized in Table 3.

## 4. Discussion

The primary goal of our investigations was to reveal the wavelength characteristics and fractal behaviours of the examined surface. Our hypothesis is presented in Figure 14, which separates three ranges by displaying the amplitude density spectrum function on a logarithmic scale. These ranges are separated from each other by the cutoff frequencies q_0_ and q_1_. 

In the literature [9,11,18,24] is an agreement on the appearance of the cutoff wavelength q_0_ that separates the constant value, the horizontal section of the PSD curve, from the linear section with a negative slope. The literature identifies this breakpoint frequency/wavelength as the dominant wavelength of the surface. This wavelength separates macrogeometry from microgeometry. We cannot make reference to roughness above this wavelength.

During our investigation of the dominant wavelength, we obtained 178 μm from the PSD function for 2D profiles (see Table 2) and 155 μm for 3D topography tests (see Figure 12). We also performed two other tests for the dominant wavelength, independent of the PSD analysis and each other. In the case of the standard *RSm* parameter, 163.1 ± 24.6 μm was obtained, while with the roughness peak identification technique, a dominant wavelength value of 161 μm was obtained. Based on the agreement of the results, it can be stated that the dominant wavelength of the examined surface is in the range of 160–180 μm. 

The rest of the experiments analysed whether micro- and nano-roughness could be separated from each other and whether bifractal/multifractal behaviours exist. The relevant results (see Table 3) are shown in Figure 15 using a novel approach.

We did not simply move from a space range to a frequency range (realized by PSD analysis), we created a frequency domain–fractal dimension link. This meant that the fractal dimension values for each frequency range were displayed.

The effect of inaccuracies caused by different measurement settings cannot be detected in the results due to the small value of the sampling step applied. Thus, the different sampling steps did not cause significant errors in the evaluation. This is supported by the same fractal results of the measurements carried out on various specimens (see purple points and blue x in Figure 15). Additionally, the effect of the measurement methodology did not influence the results because although the fractal results were significantly different in the two measurement techniques, the transition could also be observed in the largest increment in AFM measurements. In addition, there is a negligible effect of the sampling step in the case of stylus measurements as all three measurements gave the same result. Thus we can state that the above method can display the limit point q_1_ (see Figure 14) that represents the extreme value of fractality associated with machining. For our investigations, this transition is within the sampling distance of 196.1–1000 nm. This means that the fractal dimension, Df = 2.22 for machining (for micro-roughness) is only valid in the frequency range greater than 1 μm. Under 196.1 nm, the fractal dimension is around 2.5. It supposes different sources of surface unevenness in this regime than machining, and it also means that any changes in the production process will not influence this nano-roughness. 

This way we can interpret bifractal and multi-fractal terms. With the help of the sampling interval (in logarithmic scale)—fractal dimension diagram, we can identify the range of validity for each fractal dimension.

The fact that the former self-affine disappears below this range can be justified by the topographic image of the smallest AFM measurement surface. The topography shown in Figure 16 is no longer similar to the surface structures measured at larger measurement ranges (see Figure 8, Figure 9 and Figure 11). The topographic elements of the surface do not show traces of manufacturing but reflect the characteristics of material structure or other processes that determine nano-roughness.

In addition to the results presented in the article, it is important to discuss the limitations of the methodology used. In our previous work [22], we revealed the weaknesses of Fourier transform-based fractal analysis. These were: PSD analysis is sensitive to the sampling step; the value of the fractal dimension determined from the PSD curve depends significantly on the frequency range that determines the slope of the fitting line. The elimination of these negative effects can be helped by using the wavelet transformation, which has already been proven in many studies (see [36,37]) to perform better than the Fourier transformation. The wavelet transform was successfully used by Kang et al. [38] to determine the fractal dimension of rough surfaces. The wavelet analysis can represent progress compared to the presented tests in two respects: on the one hand, it can determine the fractal dimension of the surface in the micro- and nano-roughness range with greater accuracy, and on the other hand, it can more accurately determine the boundary of the micro- and nano-region, which we were able to determine only in a relatively wide range of 0.196–1 μm.

Further measurements and detailed investigations are necessary to evaluate the change in fractal dimension. One method for this is the examination of the Hurst exponent. Table 4 summarizes the Hurst exponent values for each topography.

In the literature, examination of the Hurst exponent is effectively used in several cases. Zaiser et al. [39] and Hinkle et al. [40]—in the case of deformation of metals—and Vacher et al. [41] used the Hurst exponent for compressive testing of polymers to track changes in the crystal structure. Application examples can also be found for crystallization processes in geology [42] and medicine [43]. The value of the Hurst exponent can vary between 0 and 1. A value of 0 indicates the complete absence of self-affinity. At a value of 0.5, we can talk about random changes (random walk). A Hurst exponent higher than 0.5 indicates persistence, i.e., the surface is self-affine. This can be seen in the case of macro-level roughness, which has values between 0.58 and 0.785 (see Table 4). In the examined literature, similar values were observed in the case of polycrystalline copper (see [39]) and crystalline Au and high-entropy NiCoFeTi alloy [40] as a result of forming. In the case of materials with an initial low H value (see CuZr in [40] or polymers in [41]), the process moved towards the formation of self-affinity due to the shaping effect. 

In the case of nano-roughness, the value of the Hurst exponent decreases to around 0.5 (see Table 4), thus the self-similarity changes to a random structure. A previous study [41] used polyvinyl alcohol (PVA) to create a solid polymer substrate with H about 0.3. H values around 0.4 were also detected in the case of amorphous CuZr, but values below 0.5 were found in crystallization processes (see [42,43]). Lee et al. [44] investigated the effect of anodizing on the surface structure of the aluminium 6061 material. As a result of their tests, after the various anodizing processes, the value of the Hurst exponent decreased to a value between 0.33–0.42 from the 0.57–0.65 detected in the base material. In the case of the anodized aluminium we examined, the random, non-self-affine structure of the nanostructure may also be justified, but this requires further investigation.

## 5. Conclusions

The following conclusions can be drawn from the investigations:

The dominant wavelength in the examined surface is in the range of 160–180 μm, which can be supported by multiple evaluation techniques. Because of the uneven texture of the surface, smaller (35–40 μm) and larger (220 μm) wavelength elements can be found on the surface.

For technical surfaces, the fractal dimension cannot be used for full spectrum analysis. The examined surfaces showed bifractal properties where the Df = 2.22 fractal dimensions for micro-roughness can only be interpreted in the range of 1 μm–160 μm. This result confirms the observations in [11,18,25] and limits the applicability of full-length scale tribological models such as those in [9,19,20,21].

Logarithmic sampling distance–fractal dimension diagram is suitable for determining bifractal wavelengths. This new approach provides a tool for the separation of nano- and micro-roughness. In the case of the examined surface, this cutoff wavelength is between 0.196 and 1 μm. Components with a wavelength smaller than 0.196 μm fall into the range of nano-roughness, whose fractal dimension in the examined case is Df = 2.5. 

Based on this method, further investigations can be carried out to find the sources (material structure, production parameters) of different roughness components. Surface engineers can design the micro- and nano-roughness to influence the tribological processes and to optimise the surface for operation.

## Figures and Tables

**Figure 1 materials-17-00292-f001:**
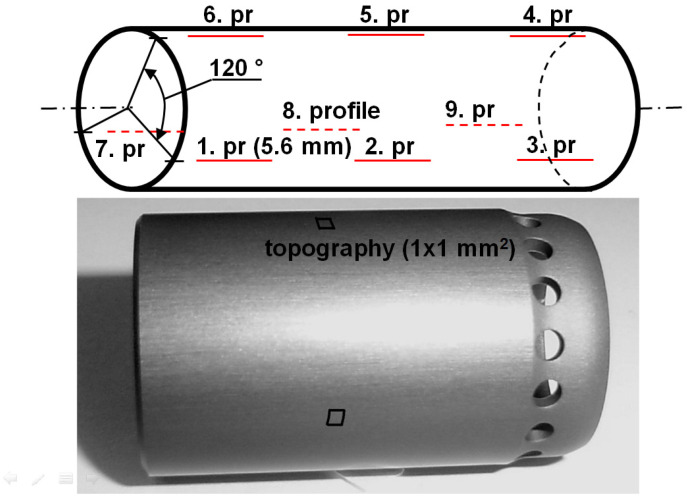
Scheme of measurements and photo of measured primary plunger. Red lines represents the positions of profile measurements on the plunger; The square represents the positions of topographic measuring areas.

**Figure 2 materials-17-00292-f002:**
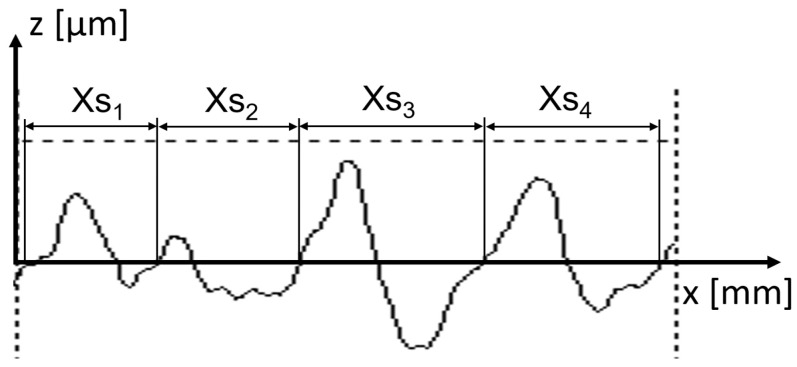
Width of profile elements.

**Figure 3 materials-17-00292-f003:**
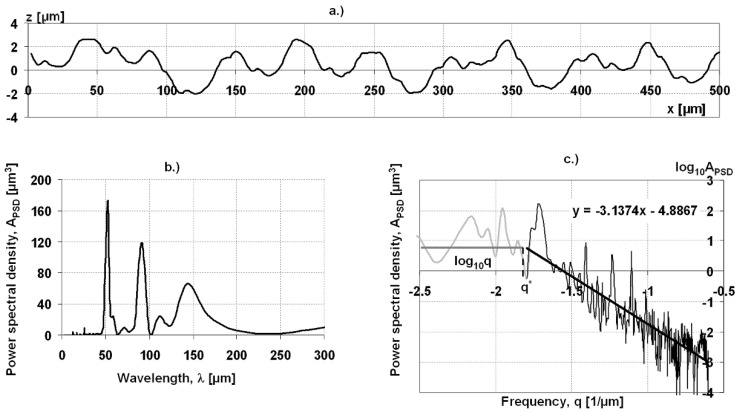
Turned profile (**a**) (feed rate is 50 µm/rev.) and PSD curve of the turned profile ((**b**) wavelength-dependent representation; (**c**) frequency-dependent representation).

**Figure 4 materials-17-00292-f004:**
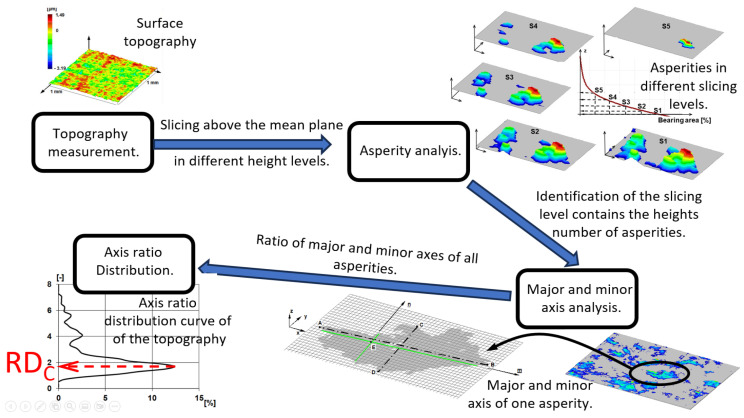
Asperity detection and axis ratio determination algorithm.

**Figure 5 materials-17-00292-f005:**
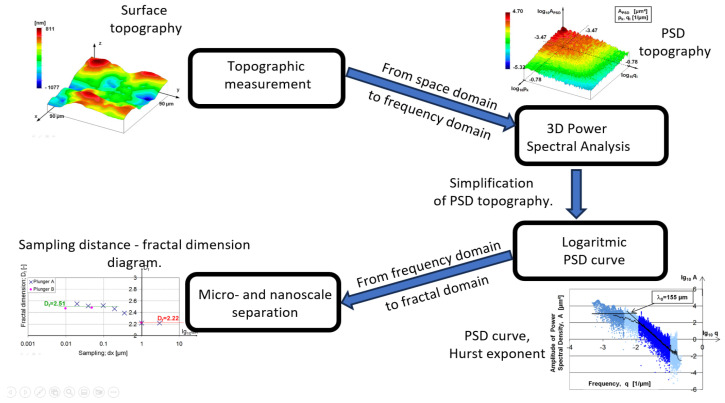
Fractal calculation from topography.

**Figure 6 materials-17-00292-f006:**
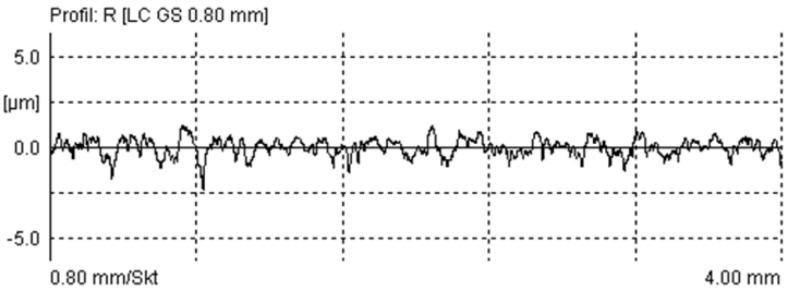
Roughness profile of measured plunger.

**Figure 7 materials-17-00292-f007:**
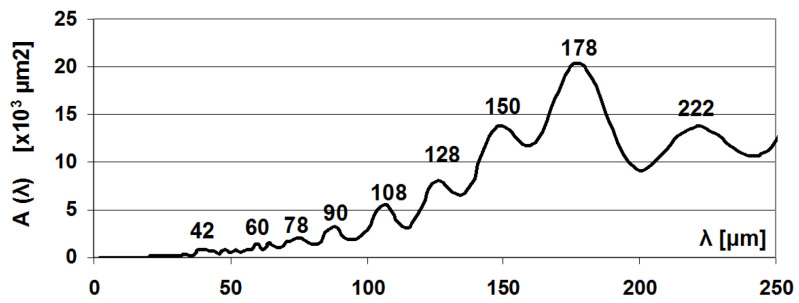
Power spectral density of measured profiles (λ, wavelength; A (λ), Amplitude of PSD).

**Figure 8 materials-17-00292-f008:**
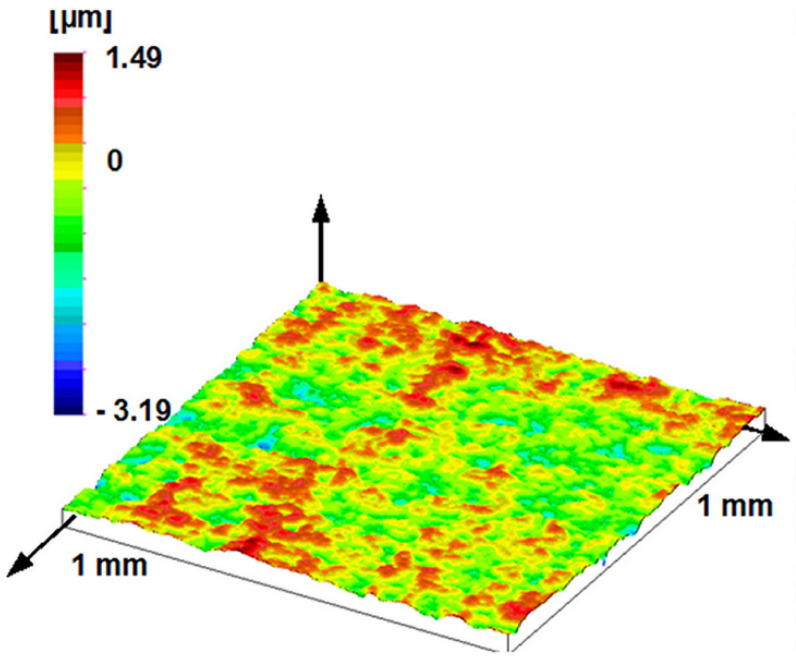
Measured topography of the plunger; stylus measurement (1 mm by 1 mm measuring area, 1 μm by 1 μm sampling).

**Figure 9 materials-17-00292-f009:**
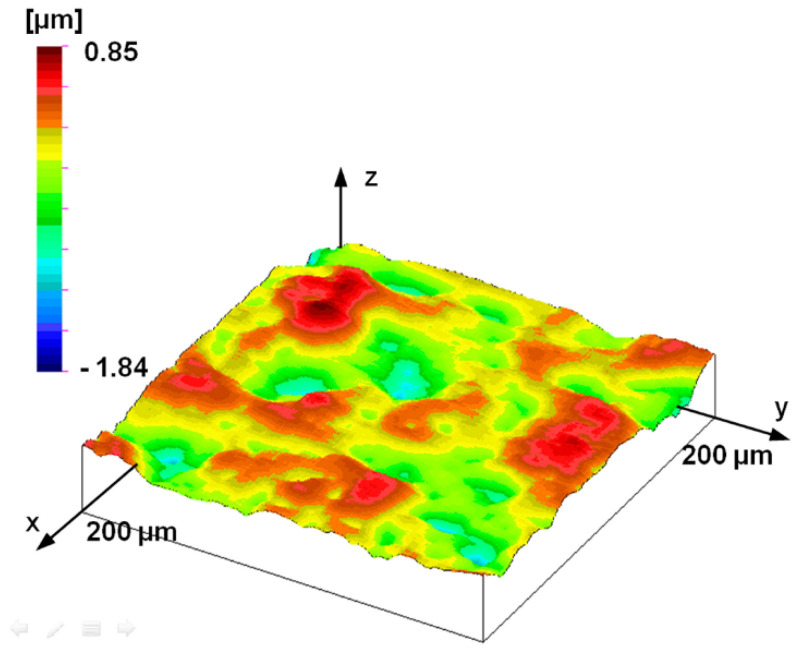
200 μm by 200 μm part of measured topography of the plunger; stylus measurement (1 mm by 1 mm measuring area, 1 μm by 1 μm sampling).

**Figure 10 materials-17-00292-f010:**
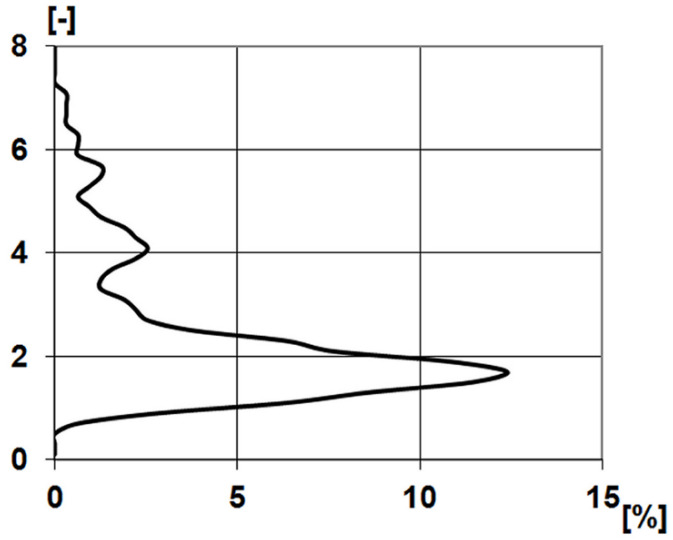
Distribution of asperity major and minor axis ratio of 3 mm by 3 mm stylus measured topography.

**Figure 11 materials-17-00292-f011:**
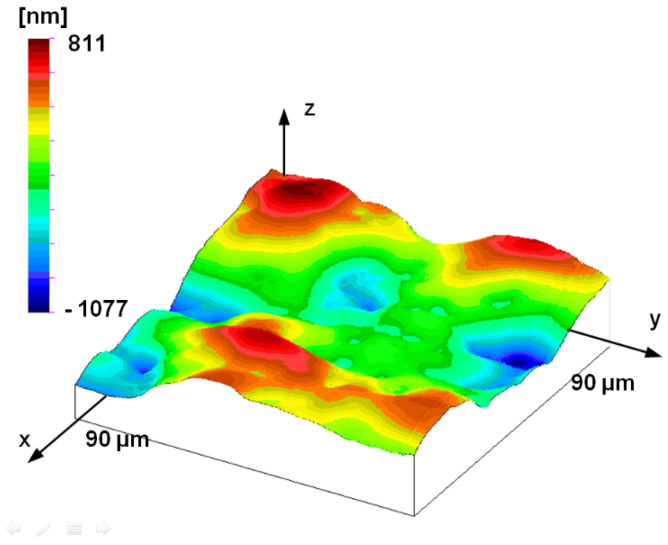
Measured topography of the plunger; AFM measurement (90 μm by 90 μm measuring area, 352.9 nm by 352.9 nm sampling).

**Figure 12 materials-17-00292-f012:**
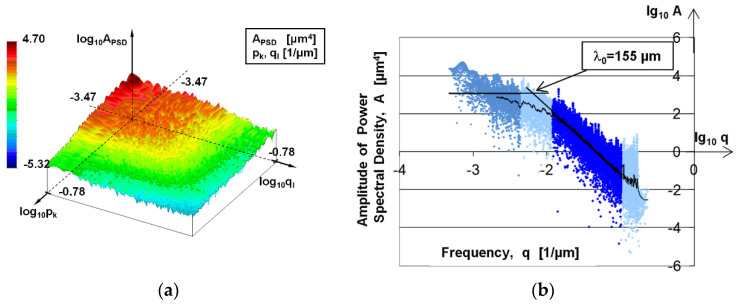
(**a**) PSD topography of 3 mm × 3 mm stylus measured surface, (**b**) PSD curve of 3 mm × 3 mm stylus measured surface.

**Figure 13 materials-17-00292-f013:**
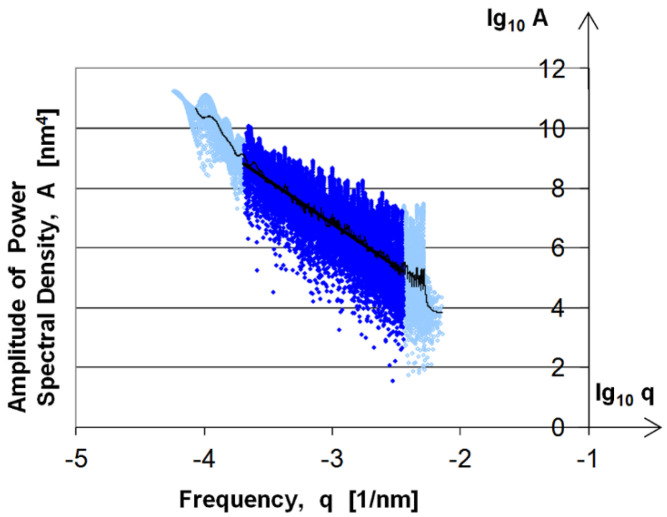
PSD curve of 25 μm × 25 μm AFM-measured surface. Light blue: PSD curve of 25 μm × 25 μm AFM-measured surface; dark blue: PSD curve on analysed frequency range.

**Figure 14 materials-17-00292-f014:**
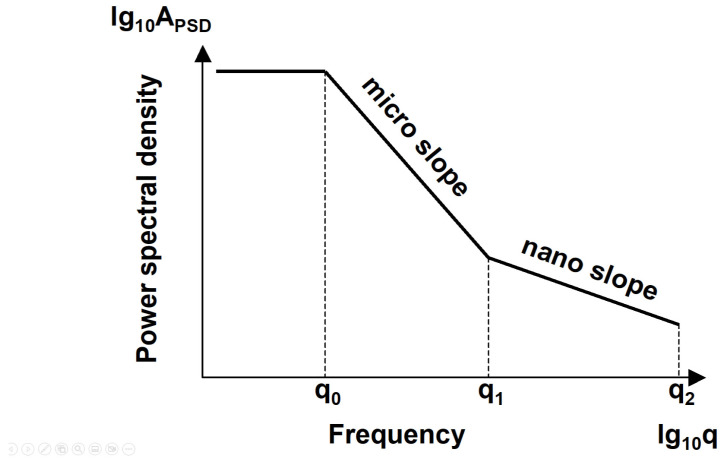
Appearance of micro- and nano-roughness on the logarithmic amplitude density spectrum.

**Figure 15 materials-17-00292-f015:**
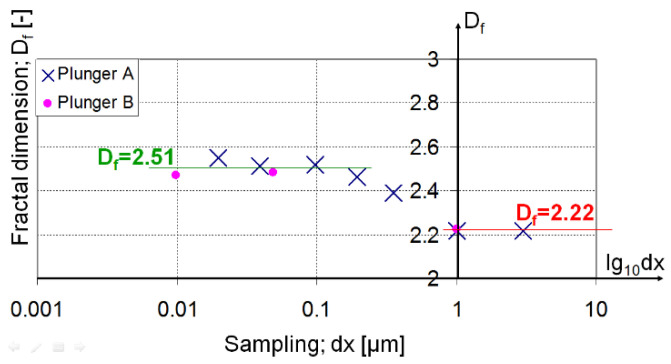
Fractal dimension sampling distance diagram.

**Figure 16 materials-17-00292-f016:**
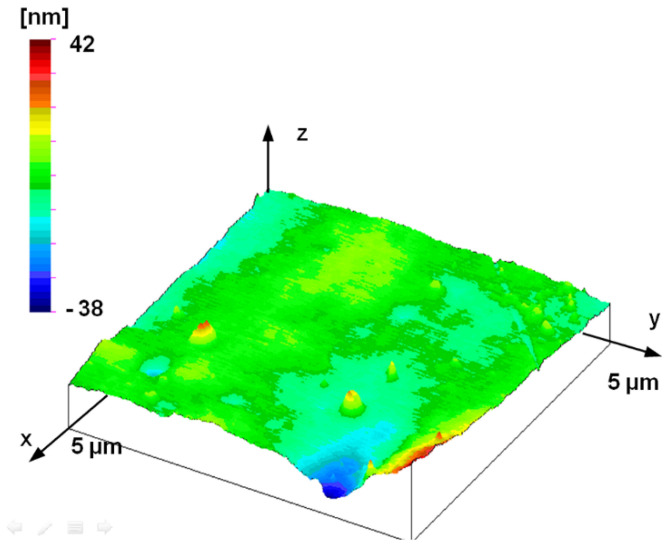
Measured topography of the plunger; AFM measurement (5 μm by 5 μm measuring area, 19.61 nm by 19.61 nm sampling).

**Table 1 materials-17-00292-t001:** Measuring conditions for topographic measurements.

	Measuring Area µm^2^	Sampling Distance nm	Number of Measurements	
			Plunger A	Plunger B
Stylus	3000 × 3000	3000	1	-
	1000 × 1000	1000	1	1
AFM	90 × 90	352.9	2	1
	50 × 50	196.1	1	1
	25 × 25	98.04	1	1
	10 × 10	39.21	1	1
	5 × 5	19.61	1	1

**Table 2 materials-17-00292-t002:** The dominant wavelength of the surface is based on different evaluation techniques.

Applied Evaluation Technique	Dominant Wavelength [μm]
*RSm* parameter (2D profile)	163.1 ± 24.6
PSD analysis (2D profile)	178
Roughness peak test	161

**Table 3 materials-17-00292-t003:** Fractal dimensions of topographies.

Measurement		Sampling Distance [μm]	Frequency Range ^1^ [1/μm]	Analysed Frequency Range ^2^ [1/μm]	Fractal Dimension	Average Fractal Dimension
Stylus	3 × 3 mm	3	0.00033–0.16	0.0118–0.1012	2.218	**2.22**
	1 × 1 mm	1	0.001–0.5	0.0164–0.2479	2.215	
AFM	90 × 90 μm	0.3529	0.0111–1.416	0.05635–0.9907	2.398	**2.39**
				0.0670–0.8239	2.356	
				0.05635–0.8239	2.420	
	50 × 50 μm	0.1961	0.02–2.549	0.1014–1.483	2.445	**2.46**
				0.1260–1.483	2.478	
	25 × 25 μm	0.0980	0.04–5.102	0.202–3.589	2.572	**2.52**
				0.252–2.967	2.458	
	10 × 10 μm	0.0392	0.1–12.75	0.507–7.415	2.552	**2.51**
				0.630–7.415	2.478	
	5 × 5 μm	0.0196	0.2–25.5	1.014–12.5	2.595	**2.55**
				1.260–12.5	2.498	

^1^ frequency range depends on the measured area and sampling distance. ^2^ analysed frequency range is the range where the line is fitted to the PSD curve.

**Table 4 materials-17-00292-t004:** Hurst exponents of topographies.

Measurement		Hurst Exponent
Stylus	3 × 3 mm	0.782
	1 × 1 mm	0.785
AFM	90 × 90 μm	0.602
		0.644
		0.580
	50 × 50 μm	0.555
		0.522
	25 × 25 μm	0.428
		0.542
	10 × 10 μm	0.448
		0.522
	5 × 5 μm	0.405
		0.502

## Data Availability

Data are contained within the article.

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
