# Peer review of "Micro- and Nano-Roughness Separation Based on Fractal Analysis"

_materials, 2024, doi:10.3390/ma17020292_

Round 1
Reviewer 1 Report (Previous Reviewer 1)
Comments and Suggestions for Authors
The authors have amended the manuscript according to suggestions. So, in my view this manuscript can be accepted in its present form.
Comments on the Quality of English Language
Some minor editing of English language is required.
Author Response
Dear Reviewer,
Thank you for taking the time to read and evaluate our manuscript and also for your positive review.
We supplemented and clarified the manuscript based on the suggestions of the reviewers in order to raise the quality of the article and make the goals and results of the research even clearer.
Sincerely,
Authors
Reviewer 2 Report (Previous Reviewer 3)
Comments and Suggestions for Authors
The manuscript has to be rejected and rewritten completely as the English is incomprehensible; furthermore, the structure of the paper is unclear, the objectives are not explained clearly and the results, discussion, and conclusions are written and presented in a very poor manner. The rigour and science cannot be judged with such poor and ineffective presentation.
Comments on the Quality of English Language
Many errors in the structure of sentences, reaching incomprehensible levels.
Author Response
Dear Reviewer,
We apologize that grammar and style problems made it impossible to review the manuscript. Thank you for taking the time to read and evaluate our manuscript, preparing the review and helping to improve it with your suggestions.
We revised and re-constructed the manuscript and supplemented it in several points.
The attached document shows the details of the revision: we marked the removed sentences in red, the new additions in blue.
We have tried our best to eliminate language problems.
We also tried to make the aims, results and conclusions of the research more understandable and clear.
Chapter 2 has been restructured in order to make the formulated goals, methodology and results easier to follow.
In addition, we expanded the methodology chapter and the discussion section by inserting new figures (Figures 2 and 14). We revised the figures showing the research methodology (Figures 4 and 5 in the current version of the manuscript) to make them even more informative.
We hope that with these changes the quality of the manuscript will meet the professional standard expected by you and the journal.
Sincerely,
Authors

Reviewer 3 Report (Previous Reviewer 4)
Comments and Suggestions for Authors
The idea of the paper is interesting and innovative. There is only a few minor suggestions.
Line 115 --- Please correct the measuring length on Fig. 1, i.e. replace comma with point.
Line 137 --- Please explain and make a notice why Plunger B was not measured with 3000 nm sampling distance, and why only Plunger A was measured twice on 352.9 nm sampling distance?
Line 201 ---- If I understood well, Figure 5 is a detail of Figure 4, so it would be nice if the position of the detail could be somehow noted on Fig. 4.
Line 267 --- What was the stylus radius and stiffness in AFM measurements?
Please add this literature in citation for improve quality of paper:
1. Effective Detection of the Machinability of Stainless Steel from the Aspect of the Roughness of the Machined Surface, M Duspara, B Savković, B Dudic, A Stoić, Coatings 13 (2), 447
2. The influence of mixing water and abrasives on the quality of machined surface, A Stoić, M Duspara, B Kosec, M Stoić, I Samardžić, Metalurgija 53 (2), 239-242

Author Response
Dear Reviewer,
Thank you for taking the time to read and evaluate our manuscript.
Thank you for your valuable suggestions, which contribute to raising the quality of the manuscript.
We supplemented and clarified the manuscript based on the suggestions of the reviewers.
We have added the literature references you suggested. Thank you for it.
We hope that with these changes the quality of the manuscript will meet the professional standard expected by you and the journal.
Sincerely,
Authors

Round 2
Reviewer 2 Report (Previous Reviewer 3)
Comments and Suggestions for Authors
The manuscript has been significantly improved. I only suggest that a Hurst exponent analysis is added so that the discussion and conclusions are improved. The Hurst exponent can be very useful in revealing underlying physical mechanisms. See e.g. M. Zaiser et al. Self-affine surface morphology of plastically deformed metals. Phys. Rev. Lett. 2004, 93 (19), 195507
Comments on the Quality of English Language
Minor editing of English language required
Author Response
Dear Reviewer,
Thank you for your valuable suggestions.
The manuscript has been supplemented with the analysis of the Hurst exponent (the changes are marked in blue). The changes appear in Chapter 2.3 and at the end of Chapter 4. For additions, we have added the following literature to the References:
- Zaiser M, Grasset F M, Koutsos V, Aifantis E C. Self-Affine Surface Morphology of Plastically Deformed Metals. Rev. Lett. 2004, 93 (19), 195507
- Hinkle A R, Nöhring, W G, Leute R, Junge T, Pastewka L. The emergence of small-scale self-affine surface roughness from deformation. Sci Adv. 2020; 6(7): eaax0847. DOI: 10.1126/sciadv.aax0847
- Vacher, R.; de Wijn, A.S. Molecular-Dynamics Simulations of the Emergence of Surface Roughness in a Polymer under Compression. Materials 2021, 14, 7327. DOI: 10.3390/ma14237327
- Dominik A, Słaby E, Śmigielski M. Hurst exponent as a tool for the description of magma field heterogeneity reflected in the geochemistry of growing crystals. Acta Geologica Polonica 2010, Vol. 60, No. 3, pp. 437–443
- Muniandy S V, Kan C S, Lim S C, Radiman S. Fractal analysis of lyotropic lamellar liquid crystal textures. Physica A 2003, Vol. 323 (1) pp. 107 – 123 DOI: 10.1016/S0378-4371(03)00026-8
- Lee W X, Farid A A, Namazi H. Electrochemical dissolution of silicon studies via noise spectroscopy. Investigation of anodised surface complexity and its correlation with surface hydrophilicity using fractal analysis. Results in Surfaces and Interfaces 2022 6., 100046
We hope that with these changes the quality of the manuscript will reach the level expected by you and the journal.
Sincerely,
Authors

This manuscript is a resubmission of an earlier submission. The following is a list of the peer review reports and author responses from that submission.
Round 1
Reviewer 1 Report
Comments and Suggestions for Authors Dear Authors, Surface roughness is considered as one of the many parameters for surface characterization of products. It appears that the authors have presented the manuscript with the objective of describing a methodology for determining the fractal range boundaries to separate micro and nano-roughness. I feel that following points need to be addressed in the manuscript. 1. The authors have tried to introduce the topic by refereeing to the available literature. It seems that the background is not clear and need to refer more recent papers in the introduction. 2. The statement of the problem to be elaborated and to be clearly stated in the manuscript. 3. Relevant papers with reference to the example taken-brake liner can be included in the introduction section. 4. Section 2.1 – Authors have explained the working of braking system in automobiles and tried to corelate this with the roughness measurement. This seems irrelevant with respect to the scope of the paper. 5. The methodology adopted is not adequately explained the manuscript. 6. Discussion about the results need to be re-written with proper justifications. 7. Conclusion part – need to improve and to be written specific quantification. What are the outcomes of the study are not stated properly. 8. Overall, the manuscript requires major modifications and content improve along with grammatical corrections.
Author Response
Dear Reviewer,
Thank you for your valuable and constructive review.
I have revised the manuscript based on your comments:
1.-3.: The introduction was extended with four paragraphs (indicated in blue) and seven new references to clarify the background and goals.
4. The working of the brake system is irrelevant with respect to the scope of the paper, but this is the industrial background of the research.
5. I have referred to three papers where the methodology is clearly presented in detail. I do not want to repeat them. However, I understand the Reviewer’s suggestion, and I extended the section with three paragraphs and three figures to present the basic concepts of methods.
6.-8. Amendments were put into Section 3-5, and a grammatical correction was carried out to improve the manuscript.
Reviewer 2 Report
Comments and Suggestions for Authors
This paper cannot be recommended in its present form. Following are the reasons from the point of review:
- The quantum of work fails to substantiate the results.
- Materials and Methods - Very basic stuffs are filled in this section. The research competence and the evaluation process are not clear
- I cant find any relevance to the content and journal scope. With two different test procedure a short experimental observations are reported in this paper. There is no reasoning, justification and solution / recommendations to the proposed problem.
- Author can give two significant reason to consider this work in Materials Journal.
Author Response
Dear Reviewer,
Thank you for your valuable and constructive review.
I have revised the manuscript based on your comments:
1.: The literature has many contradictory results in fractal, bifractal, full-length scale behaviour of surface microtopography. The quantum of our work is not enough to answer all doubtful questions, but the presented and applied method can be applied reliably in further investigations.
2. I have referred to three papers where the methodology is clearly presented in detail. I do not want to repeat them. However, I understand the Reviewer’s suggestion, and I extended the section with three paragraphs (indicated in blue) and three figures to present the basic concepts of methods.
3. The introduction was extended with four paragraphs (indicated in blue) and seven new references to clarify the background and goals. Amendments were put into Section 3-5, and a grammatical correction was carried out to improve the manuscript.
4. Two reasons to consider this work in Materials Journal:
- The manuscript is submitted to the special Issue “Friction and Wear of Materials Surfaces” (Guest Editor: Andrzej Dzierwa). One of the keywords of this special Issue is „Surface topography”. In special Issue information is mentioned: “The quality of the surface has a significant impact on the operational properties of machine elements…”, “All mechanical, physical, chemical, and geometrical aspects of the surface contact affect the surface interactions and thereby also the tribological characteristics of the system.”
- Although surface geometry and material science are different fields, in tribology the cooperation of disciplines is necessary. My new project is a good example. Plasmanitrided 42CrMo4 samples with machined, grinded, electric discharge machined, polished surfaces are under examination. The preliminary results show the followings:
* Original surface roughness highly influence the final microtopography of coated surface.
* Depending on the initial surface roughness and nitriding parameters, the coated surface roughness becomes different (in some cases finer while others rougher than the initial surface).
* In most cases, the coated surface has worse operating parameters (I mean surface roughness parameters) after nitriding.
The wear and friction test are under planning, but it seems that the optimal tribological surface will be the combination of initial geometry and nitridation. I think, in point of view of tribology, there is no material engineer or expert of surface roughness measurement, but a surface engineer who has a significant knowledge in one field and has information about the other fields of surface engineering. This special issue allows us to take a step in this direction.
Reviewer 3 Report
Comments and Suggestions for Authors
The study is of interest and generally has been conducted well, but the presentation is generally poor with numerous grammatical mistakes and typos lack of references when previous work is mentioned,. For example on top of page 7 where Bushan (sic) (please correct the misspelling) and Persson's work are mentioned. The overall style of the paper is poor. The authors need to spend considerable time to re-write the paper and get substantial help from a native speaker.
Author Response
Dear Reviewer,
Thank you for your valuable and constructive review.
I have revised the manuscript. Following changes were carried out:
- The introduction was extended with four paragraphs (indicated in blue) and seven new references to clarify the background and goals.
- I extended the section “Materials and methods” with three paragraphs and three figures to present better the basic concepts of methods.
- Amendments were put into Section 3-5, and a grammatical correction was carried out to improve the manuscript.
Reviewer 4 Report
Comments and Suggestions for Authors
- Line 57 literature 17, than line 69 literature 19…..please check chronological (order) of literature in the article, in line 90 going literature 18
- Line 92 and figure 1 - How the measurement positions are selected – by previovus investigation of authors or…..
- Line 101 – how many is the contact strength (force) between diamond stylus and material
- How many is the highest measurement area with AFM method, and how many is the smallest measurement area with stylus method
- Which type of material is made primary plunger, are these methods can be used for another type of material or for measuring roughness after cutting material where is the roughness more than 160 µm
- Please check the date of the literature, more literature are older than 5 years ago, please check these papers, recommend will be for improve the quality of citations with newest papers:
6.1 Investigation of Correlation between Image Features of Machined Surface and Surface Roughness
https://doi.org/10.17559/TV-20191212122953
6.2 The Study on Influence of the Method of Handling of Measuring Head on Measurement Results Obtained with the Use of a Portable Profilometer
https://doi.org/10.17559/TV-20160601124330
6.3 Analysis of Characteristics of Non-Commercial Software Systems for Assessing Flatness Error by Means of Minimum Zone Method
https://doi.org/10.17559/TV-20190603084835

Author Response

(The authors gave the same response as above.)

Reviewer 5 Report
Comments and Suggestions for Authors
The idea of the paper is interesting and innovative. The introduction part is informative and well organised. There is only a few minor suggestions.
- Please correct the measuring length on Fig. 1, i.e. replace comma with point.
- What was the stylus radius and stiffness in AFM measurements?
- Please explain and make a notice why Plunger B was not measured with 3000 nm sampling distance, and why only Plunger A was measured twice on 352.9 nm sampling distance?
- If I understood well, Figure 5 is a detail of Figure 4, so it would be nice if the position of the detail could be somehow noted on Fig. 4.
Author Response
Dear Reviewer,
Thank you for your valuable and constructive review.
I have revised the manuscript. Following changes were carried out:
- The introduction was extended with four paragraphs (indicated in blue) and seven new references to clarify the background and goals.
- I extended the section “Materials and methods” with three paragraphs and three figures to present better the basic concepts of methods.
- Amendments were put into Section 3-5, and a grammatical correction was carried out to improve the manuscript.
Answers for your questions and comments:
- Comma was replaced in Figure 1.
- I have no information about the details of the AFM measurement. It was carried out in a Chemical Research Centre of Hungarian Academy of Sciences, based on the relating standards and rules.
- Measurements of Plunger B was used as a check point of results. Three check points give similar result than Plunger A, and it was enough.
- The goal of Figure 4. (Fig. 9 in new version) was to get comparable result of stylus measurement and AFM measurement (Fig. 10. in new version), not to show an interesting detail of 1 mm x 1 mm stylus topography. So it was not exactly positioned.
Round 2
Reviewer 2 Report
Comments and Suggestions for Authors
Accept